# Instruction Contrastive Tuning for Zero-Shot Composed Image Retrieval

## Abstract

Composed Image Retrieval (CIR) requires retrieving a target image based on a composed query consisting of an image and accompanying text that modifies or instructs changes to the visual reference. This task is particularly challenging as it demands the model effectively follow modification instructions for accurate retrieval. Additionally, the difficulty in data acquisition hinders training models for specific tasks. To address these challenges, recent approaches explore Zero-Shot CIR (ZS-CIR), mainly leveraging CLIP-based models with tailored projections to compose images and textual modifications. However, these base models are not trained on instruction-aware data, limiting their ability to effectively combine visual and textual cues. In this paper, we propose a novel embedding method utilizing an instruction-tuned Multimodal Large Language Model (MLLM) to generate unified embeddings that seamlessly integrate images and modification instructions. Instruction-tuned MLLMs inherently align vision and text while exhibiting strong instruction-following capabilities, though they are primarily used in text generation. We introduce a two-stage training strategy to efficiently transform the MLLM's text generation capabilities into embedding extraction, and further refining its ability to follow modification instructions in CIR. Our model demonstrates significant advancements in ZS-CIR, outperforming state-of-the-art baselines across four public datasets: FashionIQ, CIRR, GeneCIS, and CIRCO. Our model highlights the potential of instruction-tuned MLLMs in capturing detailed instruction comprehension and advancing CIR systems.

## 1 Introduction

Composed Image Retrieval (CIR) is a challenging task within the field of multimodal retrieval, wherein a target image is retrieved based on a composed query consisting of both an image and accompanying text (Saito et al., 2023; Gu et al.; Agnolucci et al., 2024). The textual input typically serves as a modification or instruction applied to the visual reference, guiding the retrieval process. Such tasks are prevalent in practical applications, particularly in e-commerce scenarios (Barbany et al., 2024; Zhu et al., 2024), where users might wish to find visually similar items with slight modifications, such as a change in color or style. However, unlike conventional image-text retrieval tasks, CIR presents unique challenges in data acquisition, as it necessitates the creation of triplet data <*source image, modifier text, target image*>. This requirement significantly increases the complexity and cost of data collection, as human annotators are often needed to generate appropriate textual descriptions that link relevant images. Consequently, existing CIR datasets are limited in size and diversity, hindering the generalization ability of current models (Bai et al., 2023; Baldrati et al., 2022; Delmas et al., 2022; Ma et al., 2021; Vo et al., 2019). To address the limitations in CIR, recent research (Baldrati et al., 2023b; Ventura et al., 2024) has focused on Zero-Shot Composed Image Retrieval (ZS-CIR) as a scalable approach that can generalize to diverse contexts.

Most existing ZS-CIR models build on CLIP-based architectures (Radford et al., 2021), leveraging their robust visual-text representation capabilities. For example, Pic2Word (Saito et al., 2023) and SEARLE (Agnolucci et al., 2024) utilize lightweight projection modules to map visual embeddings into the textual space, enhancing the interaction between visual and textual modalities within CLIP's framework. Similarly, LinCIR (Gu et al.) introduces a language-only training strategy, utilizing keywords in text to represent images and circumvent the need for extensive triplet datasets. While these methods are effective, they are fundamentally constrained by the limitations of the CLIP model,

Figure 1: **Comparison of Existing CIR Approaches vs. InstructCIR.** Current state-of-the-art CIR methods typically rely on frozen VLMs, such as CLIP, which are augmented with adapters or by replacing the text encoder with a frozen LLM. These methods are constrained by the limited of instruction-following capabilities due to their frozen components. In contrast, our approach employs instruction-tuned MLLMs specifically designed for instruction-following tasks like CIR. We introduce a two-stage training strategy that more effectively adapts the model to CIR, resulting in significant improvements in flexibility and performance.

which inherently lacks the instruction-following capability needed to understand composed queries. As they are trained solely on image-text pair data, this often results in a performance bottleneck.

Recently, the advent of Large Language Models (LLMs) (Zhao et al., 2023) has opened new possibilities for improving CIR by incorporating richer instruction understanding. For instance, CIReVL (Karthik et al., 2023) leverages GPT-3.5 Brown (2020) to combine detailed image captions and textual instructions, thereby enabling a more flexible retrieval process without the need for direct model training. However, this method encounters high computational costs and rigidity during inference. VDG (Jang et al., 2024) proposes generating triplet data using a trained Multimodal LLMs (MLLMs) (Yin et al., 2023), but the MLLM itself remains peripheral to the retrieval process, limiting its direct impact on model performance. Approaches such as FROMAGe (Koh et al., 2023) and MCL (Li et al.) employ image captioning and contrastive learning to integrate LLMs with visual encoders, yet they freeze the LLMs to function purely as static encoders. As a result, these models do not fully exploit the adaptability and instruction awareness that LLMs can offer for more detailed query comprehension in ZS-CIR tasks.

Inspired by recent advances in using LLMs for text embedding benchmarks (Wang et al., 2023; Muennighoff et al., 2024), we introduce a novel embedding method based on pure instruction-tuned MLLMs for CIR in this paper. MLLMs, such as LLaVA (Liu et al., 2024c), which are trained through visual instruction tuning, offer two key advantages. First, they provide a solid vision-text alignment, crucial for multimodal tasks like CIR. Second, they are designed to follow complex instructions, a capability learned during training. However, despite their potential, MLLMs have been primarily used for text generation tasks, and their application to CIR has not been thoroughly explored. A related concurrent work (Jiang et al., 2024) trains such a model with pure text pair data. But their training does not introduce instruction awareness and is limited to the text modality, leading to sub-optimal solutions. To fully exploit the capability of MLLMs in ZS-CIR, we introduce a two-stage training strategy to adapt MLLMs for CIR. In the first stage, we perform contrastive learning (Chen et al., 2020) using pure image-text pairs to shift the MLLM's function from text generation to representation extraction, enabling it to produce multimodal embeddings suitable for retrieval. In the second stage, we enhance the MLLM's instruction-awareness by tuning it on a triplet dataset derived from an existing pair-wise dataset using GPT-4o (Achiam et al., 2023). The MLLM is trained to produce unified embeddings that align with the altered caption based on the composition of the image and query. Our approach, called **InstructCIR**, enhances model performance on ZS-CIR benchmarks, surpassing current state-of-the-art baselines. The differences between InstructCIR and prior methods are depicted in Figure 1.

In summary, our contributions are threefold: (1) To enable better image-instruction compositions, we propose an embedding strategy based on instruction-tuned MLLMs, providing superior instruction-following capabilities over previous approaches. (2) We propose a two-stage training strategy that not only transforms an MLLM's strong text generation capabilities to effective embedding extraction

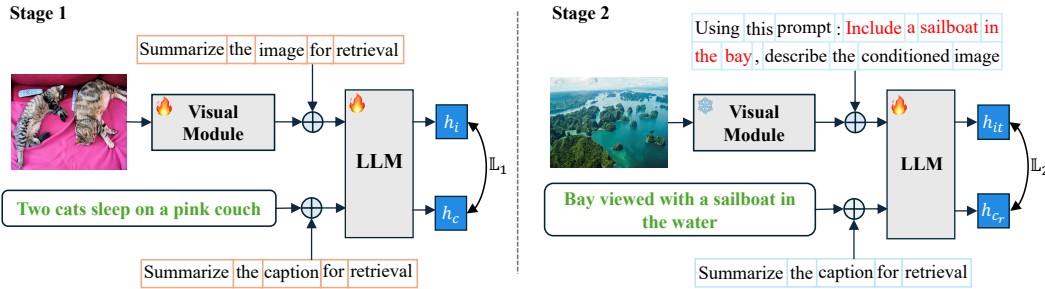

Figure 2: **The Two-Stage Training Strategy for InstructCIR.** The diagram illustrates our two-stage approach. **Stage 1**: The model is trained on image-caption pairs $(i, c)$ to align multimodal embeddings. The image is encoded by the MLLM to $h_i$, while the caption is processed to generate $h_c$. This stage establishes a shared embedding space for both modalities. **Stage 2**: The model is fine-tuned with triplet data $(i, t, c_r)$. The image and modifier text are composed into an embedding $h_{it}$, while the modified caption is encoded as $h_{c_r}$. The objective is to align $h_{it}$ and $h_{c_r}$, enhancing instruction-following abilities. The visual module includes the visual encoder and adapter. The strategy effectively handles CIR tasks by integrating visual and textual information.

but also optimally adapts it for ZS-CIR tasks. (3) Our model demonstrates significant advancements in Zero-Shot Composed Image Retrieval across four public datasets: FashionIQ, CIRR, GeneCIS, and CIRCO, outperforming existing state-of-the-arts baselines by a significant margin.

## 2 METHODOLOGY

In this section, we first outline the preliminaries of CIR and introduce the notations used in this paper. We then present InstructCIR, an MLLM-based embedding model capable of processing images, text, or a combination of both to generate a unified embedding. This unified embedding captures the composition of reference images and textual instructions, which is then used to retrieve the target images. To train this model, we propose a two-stage training strategy, as shown in Figure 2. The first stage focuses on embedding alignment, where we utilize pure image-text pairs to train the MLLM as an effective embedding model. This step is crucial for transitioning the MLLM from a text generation role to that of representation extraction, ensuring that it can generate high-quality embeddings suitable for retrieval tasks. In the second stage, we train the model to produce instruction-aware embeddings. We construct a triplet dataset <*source image, modifier text, target caption*> from an existing <*image, caption*> dataset by prompting GPT-4o to generate altering instructions and corresponding modified captions. The model is then trained using a contrastive learning approach, aiming to align the image-instruction embeddings with the target caption embeddings. This two-stage framework allows the MLLM to learn both modality alignment and instruction-following capabilities, which are essential for effective CIR.

### 2.1 PRELIMINARY

Composed Image Retrieval (CIR) involves retrieving target images based on a combination of a reference image and a modifier text which we term *instruction* or *prompt*. Formally, given a reference image $i \in \mathcal{I}$ and an instruction $t \in \mathcal{T}$ that describes the desired modification, the composed query $q = (i, t)$ is used to search for the closest target image $i_r$ within an image database $\mathcal{D} = \{i_1, i_2, \cdots, i_N\}$. The primary challenge in CIR lies in generating unified embeddings that can effectively represent the composition of both visual and textual information. Existing CIR models (Saito et al., 2023; Gu et al.; Agnolucci et al., 2024) often address this challenge by employing projection techniques on top of CLIP, where the visual representation is transformed into text tokens. These tokens, along with the instruction, are encoded into a unified embedding space. However, this approach can be limited by the modality gap that exists between visual and textual latent spaces in the joint embedding. Furthermore, the representation learning capability of these models is fundamentally constrained by the capacity of the pretrained CLIP model, which can act as a bottleneck in capturing the complex semantics required for effective CIR.

## 2.2 CONSTRUCTING INSTRUCTION-AWARE DATASET

Our training pipeline involves a two-stage strategy, where one phase is dedicated to generating instruction-aware embeddings using triplet data. In this subsection, we outline the process of constructing such a dataset. Drawing inspiration from MCL (Li et al.), we induce triplet data from the pair data available in the existing dataset CC3M (Sharma et al., 2018). Specifically, for each image-caption pair $(i, c)$, we utilize the caption $c$ to represent the image $i$. Unlike MCL, we propose leveraging GPT-4o (Achiam et al., 2023) using the Chain of Thought method (Wei et al., 2022). This involves providing GPT with the original caption $c$ and two few-shot examples. In each example, GPT brainstorms an instruction by first identifying key concepts in the caption, then deriving a modified caption based on the instruction. For instance, given the caption "A husky is lying on the grass," we identify the object husky, the action lying, and the background grass. By changing the action to running, the modified caption becomes "A husky is running on the grass." The query caption $c$ is given to GPT followed by examples, resulting in a triplet $(i, t, c_r)$ where $t$ is the brainstormed instruction and $c_r$ is the modified caption. Due to space limitations, figures illustrating the processing pipeline and difference from the MCL data processing are included in Appendix B.

Notably, acquiring the modified image $i_r$ is often more complex and costly. Therefore, we use the constructed triplet $(i, t, c_r)$ directly for model training. Since the ultimate retrieval target in CIR is an image, rather than text, we propose aligning the joint embedding space of images and text in the first stage. This alignment ensures that, when the model is trained to retrieve the modified caption in the second stage, the resulting embeddings are consistent with those of the modified images. This approach facilitates effective training for CIR by aligning textual modifications with visual changes.

## 2.3 INSTRUCTION-AWARE CONTRASTIVE LEARNING

**Model Architecture.** We utilize the MLLM as the embedding model due to its ability to generate unified embeddings for both images and text. In common MLLMs, images are first processed by the visual encoder, such as a Vision Transformer (ViT) (Alexey, 2020). The resulting patch embeddings are then projected into the LLM embedding space via an adapter, allowing them to be concatenated with the input text embeddings. The concatenated sequence is subsequently fed into the LLM component to produce the final output. When only textual input, such as captions and prompts, is provided, it is directly tokenized and processed by the LLM, bypassing the visual encoder. To extract a comprehensive embedding from the MLLM, we append a special token [EOS] at the end of the input sequence, ensuring that the model's final output captures the entire context. The input sequence, including this [EOS] token, is forwarded through the model, and the embedding corresponding to the [EOS] token in the output sequence is used as the global representation $h$. This forward process is illustrated in Figure 3. We use subscripts to denote representations from different inputs in later sections.

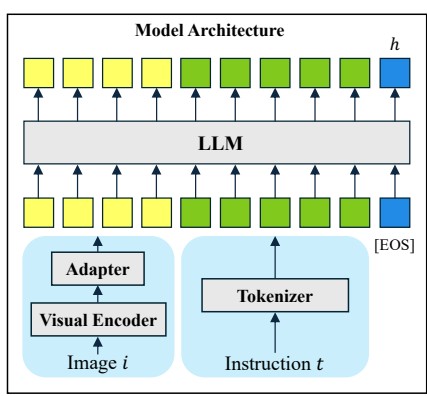

Figure 3: **Model Architecture**: The image $i$ is processed by a visual encoder and adapter, while the instruction $t$ is tokenized. Both are concatenated and fed into the LLM along with the [EOS] token. The final output at the [EOS] token provides the unified embedding $h$.

**Embedding Alignment.** Although MLLMs have achieved alignment between vision and text inputs, they are typically not optimized for embedding extraction. In the first stage of our training process, we aim to learn a joint retrieval embedding space tailored for multimodal inputs. Specifically, we employ an image-caption pair dataset $(i, c) \in \mathcal{D}_1$ for contrastive learning. Similar to Jiang et al. (2023) for text embedding extraction, a unified instruction is used to prompt the model to summarize each image or text, formulated as: *Summarize the image (text) in one word:*. Both the image and the text, along with their respective instructions, are then fed into the model to obtain the embeddings $(h_i, h_c)$. The model is trained using an InfoNCE loss (Oord et al., 2018), as defined

in Equation 1. During this stage, all components of the MLLM, including the visual encoder, the adapter, and the LLM, are trainable. This comprehensive training approach is designed to facilitate the learning of a unified embedding space.

$$\mathbb{L}_1 = -\frac{1}{2}\left(\log\frac{\phi(h_i, h_c)}{\phi(h_i, h_c) + \sum_{n\in\mathbb{N}_1}\phi(h_i, h_{n_c})} + \log\frac{\phi(h_c, h_i)}{\phi(h_c, h_i) + \sum_{n\in\mathbb{N}_1}\phi(h_c, h_{n_i})}\right) \quad (1)$$

Here, $\phi(h_i, h_c) = \exp\left(\frac{1}{\tau}\cos(h_i, h_c)\right)$ represents the scaled cosine similarity, where $\tau$ is the temperature parameter. $\mathbb{N}_1$ denotes the set of negative samples for the current batch, and $h_{n_i}$ ($h_{n_c}$) refers to negative image (caption) correspondences. We utilize in-batch samples as well as hard negative samples (if exist) to construct the negative set. Details are shown in the **left** part of Figure 2.

**Instruction Contrastive Tuning.** From the previous stage, we have obtained an MLLM-based embedding model capable of encoding both images and text into a joint embedding space. In this stage, our objective is to train the model to generate instruction-aware embeddings that generalize effectively to different instructions. To enhance the model's zero-shot performance on unseen composed image retrieval, we incorporate two distinct prompt templates during training.

Given the generated triplet data $(i, t, c_r) \in \mathcal{D}_2$, we use the first template to integrate the modification instruction $t$. This template, such as *"Using this prompt: {}, describe the conditioned image: "*, is sampled from a predefined set and is designed to guide the model in understanding how the image should be modified according to the instruction. The reference image $i$ and the formatted instruction are encoded by the model into a composed embedding $h_{it}$.

The second template is employed to guide the model in retrieving the modified caption. We use a summary prompt, such as *"Summarize the image for retrieval: "*, sampled from another predefined set, to encode the modified caption $c_r$. This template helps the model learn to distill key information into a retrieval-friendly representation. The model encodes the prompt and the modified caption to generate the embedding $h_{c_r}$. Details of the prompt template sets are provided in Appendix C.

By using two different templates, we encourage the model to distinguish between the task of understanding modification instructions and the task of generating embeddings that are optimal for retrieval. This distinction is crucial for enhancing the model's ability to generalize to unseen data in a zero-shot setting. Finally, we compute the InfoNCE loss between the composed embedding $h_{it}$ and the target embedding $h_{c_r}$ as shown in Equation 2. During this stage, the visual encoder and adapter are frozen, and only the LLM is trained to refine its instruction-following capabilities. Details of this stage is shown in the **right** part of Figure 2.

$$\mathbb{L}_2 = -\log\frac{\phi(h_{it}, h_{c_r})}{\phi(h_{it}, h_{c_r}) + \sum_{n\in\mathbb{N}_2}\phi(h_{it}, h_n)} \quad (2)$$

Here, $\phi$ represents the scaled cosine similarity. The negative set $\mathbb{N}_2$ consists of other in-batch modified captions and the original caption $c$ of the current sample, serving as a hard negative.

## 3 EXPERIMENTS

### 3.1 SETTINGS

For our experiments, we adopt the xtuner/llava-phi-3-mini-hf (Contributors, 2023) as the base model for InstructCIR, chosen for two key reasons: (1) LLaVA-based models (Liu et al., 2024c) represent a widely-used paradigm in current MLLMs, and testing on such a model provides valuable insights that can be generalized to similar architectures. (2) Computational efficiency is a significant concern with general LLMs due to their high memory demands. By utilizing a lightweight, on-device variant like Phi-3-mini (Abdin et al., 2024), we mitigate these computational challenges. To ensure consistency with the baseline models, we do not directly apply the checkpoint from xtuner/llava-phi-3-mini-hf. Instead, we re-train a variant, denoted as LLaVA-Phi, by modifying the visual encoder from the original openai/clip-vit-large-patch14-336 (which uses a $336 \times 336$ resolution) to

| Method | CIRR | | | | | | CIRCO | | | |
|---|---|---|---|---|---|---|---|---|---|---|
| | R@1 | R@5 | R@10 | $R_s$@1 | $R_s$@2 | $R_s$@3 | mAP@5 | mAP@10 | mAP@25 | mAP@50 |
| Pic2word | 23.90 | 51.70 | 65.30 | 53.76 | 74.46 | 87.08 | 8.72 | 9.51 | 10.64 | 11.29 |
| SEARLE | 24.24 | 52.48 | 66.29 | 53.76 | 75.01 | 88.19 | 11.68 | 12.73 | 14.33 | 15.12 |
| KEDs | 26.40 | 54.80 | 67.20 | - | - | - | - | - | - | - |
| Context-I2W | 25.60 | 55.10 | 68.50 | - | - | - | - | - | - | - |
| CIReVL | 24.55 | 52.31 | 64.92 | 59.54 | 79.88 | 89.69 | 18.57 | 19.01 | 20.89 | 21.80 |
| LinCIR | 25.04 | 53.25 | 66.68 | 57.11 | 77.37 | 88.89 | 12.59 | 13.58 | 15.00 | 15.85 |
| FROMAGe | 10.96 | 31.40 | - | 34.07 | - | - | 4.00 | 4.44 | 5.26 | 5.73 |
| MCL | 26.22 | 56.84 | - | 61.45 | - | - | 17.67 | 18.86 | 20.80 | 21.68 |
| **InstructCIR** | **35.18** | **65.12** | **77.61** | **67.54** | **84.77** | **93.61** | **22.32** | **23.80** | **26.25** | **27.32** |

Table 1: **Comparison of Zero-Shot CIR Models on CIRCO and CIRR Test Sets.** Baseline results are directly taken from original papers. Results not reported are marked as "-". Our model significantly outperforms baseline ZS-CIR models across various metrics and datasets.

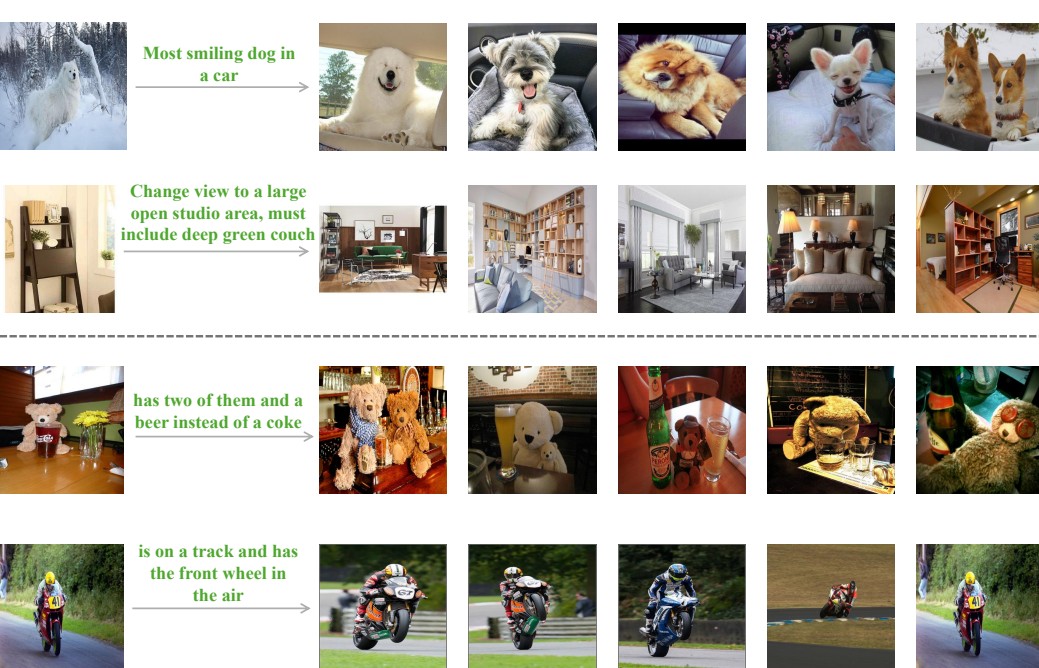

Figure 4: Examples from **CIRR** (top) and **CIRCO** (bottom) validation sets. Results are ranked from highest (left) to lowest (right) similarity. InstructCIR effectively retrieves images across a wide variety of modifier instructions from source images.

openai/clip-vit-large-patch14 with a $224 \times 224$ resolution. Additionally, we upgrade the LLM component to the latest Phi-3.5-mini. *Note that the goal of training such a variant is solely to make our experiments consistent with the baselines.* In ablation studies, we show our training strategy can be directly applied to existing MLLMs, such as microsoft/Phi-3.5-vision-instruct (Abdin et al., 2024), and explore cutting-edge techniques like dynamic high-resolution for CIR tasks.

For the first stage of training, we utilize two image-caption datasets: LLaVA-Pretrain (Liu et al., 2024a) and FOIL (Shekhar et al., 2017), an extension of the MSCOCO 2014 dataset (Lin et al., 2014), where each image-caption pair includes hard negative captions to enhance learning. In the second stage, we derive a 2M triplet dataset from the CC3M, termed **CC3M-Instruct**. The more data details are provided in Appendix B.2. We randomly select a 300K subset from CC3M-Instruct, as it provides efficient training without loss in performance compared to the full dataset. The impact of training data scale is explored in the ablation studies. Both stages are trained for one epoch. To optimize efficiency, we employ LoRA (Hu et al., 2021)and DeepSpeed ZeRO-2 (Rajbhandari et al., 2020) during training. The first stage takes approximately 1.5 hours, while the second stage requires around 2.5 hours on a cluster of four H100 GPUs. More hyperparameters and configuration details

| Method | Shirt | | Dress | | Toptee | | Average | |
|---|---|---|---|---|---|---|---|---|
| | R@10 | R@50 | R@10 | R@50 | R@10 | R@50 | R@10 | R@50 |
| Pic2word | 26.20 | 43.60 | 20.0 | 40.2 | 27.90 | 47.40 | 24.70 | 43.70 |
| SEARLE | 26.89 | 45.58 | 20.48 | 43.13 | 29.32 | 49.97 | 25.56 | 46.23 |
| KEDs | 28.90 | 48.00 | 21.70 | 43.80 | 29.90 | 51.90 | 26.80 | 47.90 |
| Context-I2W | 29.70 | 48.60 | 23.10 | 45.30 | 30.60 | 52.90 | 27.80 | 48.90 |
| CIReVL | 29.49 | 47.40 | 24.79 | 44.76 | 31.36 | 53.65 | 28.55 | 48.57 |
| LinCIR | 29.10 | 46.81 | 20.92 | 42.44 | 28.81 | 50.18 | 26.28 | 46.49 |
| **InstructCIR** | **30.96** | **50.10** | **25.11** | **46.18** | **32.32** | **54.22** | **29.46** | **50.16** |

Table 2: **Comparison of Zero-Shot CIR Models on FashionIQ**. Our model significantly outperforms state-of-the-art methods across all FashionIQ sub-benchmarks.

are included in Appendix D. All codes, processed datasets, and model checkpoints will be released to the public to ensure reproducibility.

## 3.2 DATASETS AND BASELINES

We evaluate our model using four well-established zero-shot CIR benchmarks: FashionIQ (Wu et al., 2021), CIRR (Liu et al., 2021a), CIRCO (Baldrati et al., 2023a), and GeneCIS (Vaze et al., 2023). While FashionIQ is an early benchmark for CIR, its domain is restricted to fashion e-commerce images. In contrast, CIRR and CIRCO focus on more general natural images. CIRR is the first CIR dataset centered on natural images, but it suffers from the limitation of having only one target image per query, leading to potential false negatives. On the other hand, CIRCO improves upon this by providing multiple target images per query, which reduces the likelihood of false negatives and offers a more comprehensive evaluation of retrieval accuracy. GeneCIS is a dataset for conditional image retrieval. It defines four types of conditions as focusing or changing attributes or objects in images. In line with common practice, we report Recall@k ($R@k$) for FashionIQ, CIRR, and GeneCIS, with an additional subset metric for CIRR denoted as $R_s@k$. For CIRCO, where multiple correct images can correspond to a single query, we use mean Average Precision ($mAP@k$) to capture both precision and recall across different retrieval positions. Importantly, although some benchmarks, specifically CIRCO and GeneCIS, utilize images from MSCOCO, they are sourced from different versions and dataset splits with FOIL (2017 vs. 2014, and Unlabeled/Validation vs. Training sets). Moreover, aside from the images, FOIL does not include the modifier texts and corresponding targets from these benchmarks but contains only captions. Therefore, there is no overlap between our training and testing settings. We also report full results using only LLaVA-Pretrain in the first stage in Appendix E. Note that CIRR and CIRCO have hidden test sets accessible only through server submissions. We report the main results on these test sets following baseline protocols but conduct ablations on the corresponding validation sets except Section 3.4.3.

We compare our approach against state-of-the-art CIR models, focusing on those that use ViT-L ($224 \times 224$) as the visual backbone. These baselines can be divided into two primary categories: (1) CLIP-based models, including Pic2Word (Saito et al., 2023), CIReVL (Karthik et al., 2023), Context-I2W (Tang et al., 2024), KEDs (Suo et al., 2024), SEARLE (Agnolucci et al., 2024), and LinCIR (Gu et al.); and (2) LLM-based models, such as FROMAGe (Koh et al., 2023) and MCL (Li et al.). We exclude baselines (Zhang et al., 2024) that rely on proprietary data, as their distribution and overlap with public benchmarks are unknown.

## 3.3 MAIN RESULTS

For the CIRCO benchmark, Table 1 reports performance on the hidden test set, which is accessible via the submission server provided by Baldrati et al. (2023b). Our approach demonstrates substantial improvements over existing methods, such as Pic2Word and SEARLE, achieving an $mAP@5$ of 22.32%. This represents a notable increase of 13.60% over Pic2Word and 10.64% over SEARLE. Additionally, when compared to the strongest baseline, CIReVL— which leverages BLIP-2 (partially trained on MSCOCO 2014) and GPT-3.5-turbo—our model achieves an improvement of 4.79% in $mAP@5$. These results are particularly significant given that CIRCO is the most rigorously annotated dataset in the CIR field. Unlike other datasets, CIRCO incorporates multiple correct target images for each query, addressing the inherent ambiguity of the CIR task, where tex-

| Method | Focus Attribute | | | Change Attribute | | | Focus Object | | | Change Object | | | Average |
|--------|------|------|------|------|------|------|------|------|------|------|------|------|------|
| | R@1 | R@2 | R@3 | R@1 | R@2 | R@3 | R@1 | R@2 | R@3 | R@1 | R@2 | R@3 | R@1 |
| Pic2Word | 15.65 | 28.16 | 38.65 | 13.87 | 24.67 | 33.05 | 8.42 | 18.01 | 25.77 | 6.68 | 15.05 | 24.03 | 11.15 |
| SEARLE | 17.10 | 29.60 | 40.70 | **16.30** | 25.20 | 34.20 | 12.00 | 22.20 | 30.90 | 12.00 | 24.10 | 33.90 | 14.35 |
| LinCIR | 16.90 | 29.95 | 41.45 | 16.19 | 27.98 | 36.84 | 8.27 | 17.40 | 26.22 | 7.40 | 15.71 | 25.00 | 12.19 |
| CIReVL | 19.50 | 31.80 | 42.00 | 14.40 | 26.00 | 35.20 | 12.30 | 21.80 | 30.50 | **17.20** | 28.90 | 37.60 | 15.85 |
| **InstructCIR** | **21.25** | **34.55** | **46.85** | 16.15 | **28.74** | **39.73** | **17.55** | **28.01** | **36.94** | 17.04 | **28.98** | **37.70** | **18.00** |

Table 3: **Comparison of Zero-Shot CIR Models on GeneCIS**. Our model shows superior performance over state-of-the-art methods across four different scenarios of GeneCIS.

tual modifications of an image can yield multiple valid outcomes. The strong performance of our model on this dataset provides key evidence of its robustness and ability to handle complex retrieval tasks with greater precision than current state-of-the-art methods.

For the CIRR dataset, the results from the hidden test set, returned by the submission server as in Liu et al. (2021b), are presented in Table 1. CIRR presents unique challenges due to its noisy nature, where the modifying instruction plays a much larger role in the retrieval process, while the reference image often has less direct correlation with the target image. Despite this noise, our model achieves substantial improvements, surpassing Pic2Word and SEARLE by 11.28% and 10.94% in $R@1$, respectively. Among the baselines, the most competitive result comes from MCL, an LLM-based model also trained on triplet data. However, our model surpasses MCL by 8.96% in $R@1$ and 6.09% in $R_s@1$, underscoring the effectiveness and flexibility of our approach in handling complex CIR tasks where the relationship between images and instructions is ambiguous.

Figure 4 visualizes retrieval examples from CIRR and CIRCO with instructions impacting different semantic elements of the reference image such as viewpoint, layouts, object counts, poses, and background changes. This provides further indication about the diverse applicability of our setup.

For the FashionIQ dataset, Table 2 highlights the performance of our model compared to previous zero-shot methods. Our model achieves impressive improvements, with 4.76% and 3.9% increases in average $R@10$ over Pic2Word and SEARLE, respectively. It is important to note that our training data primarily consists of natural images, whereas FashionIQ is a domain-specific dataset focused on fashion e-commerce images. This significant performance on FashionIQ demonstrates the strong generalization capability of our model, which can effectively transfer knowledge from natural image domains to more specialized image retrieval tasks. These results illustrate the proficiency of our model in addressing the diverse challenges posed by both fashion-specific and general natural image datasets in zero-shot settings.

For the GeneCIS dataset, Table 3 demonstrates the superiority of our model. It surpasses Pic2Word and SEARLE by 61.43% and 20.27% in average $R@1$, and outperforms all baselines in $R@2$ and $R@3$, demonstrating its outstanding capability in processing conditional image retrieval.

### 3.4 ABLATIONS

Our ablation studies aim to address the following key questions regarding the effectiveness and robustness of our proposed method: **Q1:** How do different training stages contribute to model performance? **Q2:** What is the impact of training data on model effectiveness? **Q3:** Can our approach be easily adapted to sophisticated MLLM mechanisms?

#### 3.4.1 Q1: HOW DO DIFFERENT TRAINING STAGES CONTRIBUTE TO MODEL PERFORMANCE?

To assess the impact of each stage in our training strategy, we conducted ablation studies, isolating the contributions of Stage 1 and Stage 2. As presented in Table 4, the combination of both stages consistently yields superior performance, with the second stage contributing more significantly. Stage 1 establishes a robust joint embedding space for images and text through contrastive learning on image-caption pairs. Though not directly related to CIR, it reduces the modality gap, which is crucial for handling complex compositional queries in Stage 2. In contrast, Stage 2 directly aligns the model's training objective with the CIR task by using triplet-based contrastive learning. Here, the model is explicitly trained to match the image-modification pair to the modified caption,

| Stage 1 | | Stage 2 | CIRCO | CIRR | FashionIQ | Avg. |
|---|---|---|---|---|---|---|
| LLaVA-Pretrain | FOIL | CC3M-Instruct | mAP@5 | R@10 | R@10 | |
| ✓ | ✗ | ✗ | 5.10 | 46.90 | 19.91 | 23.97 |
| ✗ | ✓ | ✗ | 5.22 | 47.12 | 19.13 | 23.82 |
| ✓ | ✓ | ✗ | 5.92 | 50.25 | 23.16 | 26.44 |
| ✗ | ✗ | ✓ | 19.43 | 73.20 | 24.19 | 38.94 |
| ✓ | ✗ | ✓ | 20.10 | 75.80 | 28.90 | 41.60 |
| ✗ | ✓ | ✓ | 20.43 | **77.04** | 26.09 | 41.18 |
| ✓ | ✓ | ✓ | **21.27** | 76.23 | **29.46** | **42.32** |

Table 4: **Results of different stages.** LLaVA-Pretrain and/or FOIL are used in the first stage, which contain image-caption pairs. The triplet dataset CC3M-Instruct is used in the second stage. **Bold** indicates the highest scores and Underline indicates the second highest scores.

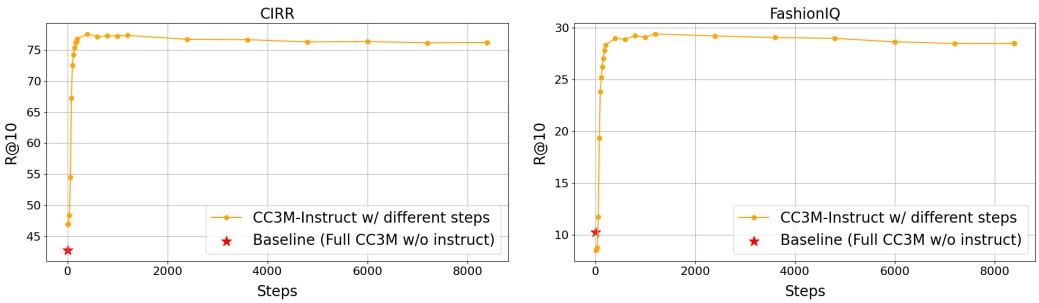

Figure 5: **Effectiveness of the triplet data by scale.** The baseline is our model trained with the whole original CC3M pair data. The plot demonstrates the performance curve on validation sets by steps. We see that the performance improves rapidly at beginning steps.

which mirrors the actual CIR task during inference. This stage fine-tunes the model to follow modification instructions and adapt its embeddings accordingly. By directly optimizing for the target task, Stage 2 has a more substantial influence on final performance. We observe that Stage 2 alone, without the pre-alignment from Stage 1, performs suboptimally, indicating that the initial feature alignment plays a critical supporting role. This interplay between stages highlights the importance of a progressive learning strategy that first handles modality discrepancies before transitioning to task-specific fine-tuning. Additionally, the combination of LLaVA-Pretrain and FOIL in the first stage performs better than using either dataset alone, emphasizing the importance of exposing the model to diverse data during feature alignment. We also observe that the larger scale of LLaVA-Pretrain (560K samples) outperforms FOIL (60K samples), likely due to the data scale advantage.

### 3.4.2 What is the impact of training data on model effectiveness?

To evaluate the effectiveness of the triplet dataset and the training scale, we conducted experiments using different dataset sizes of the CC3M-Instruct and the original pair-wise CC3M datasets. Figure 5 shows the performance across varying training steps. We observe that using the entire original pair data yields results similar to those obtained in the first-stage datasets, whereas the use of triplet data significantly improves performance. The recall grows rapidly up to around 1200 steps (approximately 300K triplets), after which it stabilizes. Continued training introduces fluctuations and potential overfitting. This pattern suggests that MLLMs quickly adapt to the training data, emphasizing the importance of carefully managing training data scale. We find that a 300K subset balances efficiency and performance, and recommend using diverse, regular-sized datasets for future training.

In the second stage of training, we use the original caption $c$ as the hard negative of a triplet $(i, t, c_r)$. In Table 5, we show the effectiveness of incorporating hard negatives. It can be observed that the incorporation of hard negatives improve the performance because the modified caption and original caption may look similar and contrasting them in training can enhance the model ability to understand the difference. In addition, the first row in the table shows the opposite strategy that uses the original image-caption pair with the modified caption as the hard negative. Results again signify the

effectiveness of training with the modification query and modified caption against the original pair. Furthermore, in the second training stage, we utilize randomly selected prompt templates, whereas row 3 demonstrates the opposite approach by using fixed prompts for training. The results reveal the necessity of employing diverse templates.

| Hard Neg. & Template Strategy | CIRR | | | FashionIQ | |
|---|---|---|---|---|---|
| | R@1 | R@5 | R@10 | R@10 | R@50 |
| $(i, c)$, $c_r$ as hard neg. | 10.79 | 33.10 | 46.04 | 8.12 | 19.20 |
| $(i, t, c_r)$ w/o hard neg. | 33.08 | 63.98 | 75.58 | 27.29 | 48.81 |
| Fixed templates | 33.60 | 63.81 | 74.85 | 27.10 | 47.53 |
| Ours | **34.63** | **64.90** | **76.23** | **29.46** | **50.16** |

Table 5: **Results of Different Hard Negative and Template Strategies.** "Ours" denotes the use of $(i, t, c_r)$ triplets with $c$ as hard negatives and randomly selected templates during training, as opposed to fixed templates. $(i, c)$ indicates using the original image-caption pairs.

### 3.4.3 CAN OUR APPROACH BE EASILY ADAPTED TO SOPHISTICATED MLLM MECHANISMS?

In this section, we analyze sophisticated MLLM mechanisms with a latest MLLM microsoft/Phi-3.5-vision-instruct on our training strategies. The difference between microsoft/Phi-3.5-vision-instruct and xtuner/llava-phi-3-mini-hf are two folds: (1) The former is trained with three stages including the feature alignment, instruction tuning, and preference optimization (Rafailov et al., 2024) while the latter is only trained with the first two stages; (2) Phi-3.5-vision leverages the dynamic high resolution (Liu et al., 2024b; Dong et al., 2024). An input image that is oversize will not only be resized but also chunked into several parts. The resized image and image parts will be encoded by the visual encoder and fed to the LLM together. While such an operation is powerful, it also suffers from higher computational cost in both training and inference as more patches are fed to the LLM.

| Method | CIRR | | | | | | CIRCO | | | |
|---|---|---|---|---|---|---|---|---|---|---|
| | R@1 | R@5 | R@10 | $R_s$@1 | $R_s$@2 | $R_s$@3 | mAP@5 | mAP@10 | mAP@25 | mAP@50 |
| E5-V | 34.17 | 64.39 | 75.98 | 64.02 | 83.35 | 92.70 | 20.44 | 22.06 | 24.25 | 25.26 |
| **InstructCIR** | 35.18 | 65.12 | 77.61 | 67.54 | 84.77 | 93.61 | 22.32 | 23.80 | 26.25 | 27.32 |
| **InstructCIR+** | **37.93** | **69.40** | **80.36** | **70.72** | **87.18** | **94.34** | **26.12** | **27.18** | **29.50** | **30.53** |

Table 6: **Results of sophisticated MLLMs on CIRR and CIRCO test sets**. InstructCIR uses LLaVA-Phi as the base model, consistent with the main experiments, while InstructCIR+ uses microsoft/Phi-3.5-vision-instruct as the base model.

We use the microsoft/Phi-3.5-vision-instruct model as the base to conduct ablations on the CIRR and CIRCO *test sets*, referring to this variant as **InstructCIR+**. We compare it with a concurrent work, E5-V (Jiang et al., 2024), which utilizes LLaVA-NeXT (Liu et al., 2024b; 2023) as the backbone—a twice larger MLLM equipped with dynamic high resolution. Our method differs from E5-V in that our training strategy is multimodal and instruction-aware, whereas E5-V trains the MLLM only on pure text pair data. Results are shown in Table 6. As observed, Phi-3.5-Vision improves upon LLaVA-Phi despite both using Phi-3.5-mini as LLMs. These findings indicate that these techniques can benefit CIR and that our training strategy can be directly applied to existing MLLMs. Notably, both InstructCIR and InstructCIR+ outperform E5-V, even without LLaVA-Phi using dynamic high resolution, highlighting the effectiveness of our instruction-aware training strategy.

## 4 CONCLUSION

In this paper, we present InstructCIR, a ZS-CIR model built on instruction-tuned MLLMs. Our approach highlights the potential of MLLMs in CIR systems, leveraging their robust instruction-following abilities and strong vision-language alignment to address the lack of instruction-awareness in previous methods. The proposed two-stage training strategy effectively refines the MLLM's text generation capabilities for embedding extraction while enhancing its instruction-following within the CIR context. We believe this work provides valuable insights into model selection and training strategies, paving the way for future advancements in ZS-CIR.

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

## A    RELATED WORKS

### A.1    INSTRUCTION TUNING

Instruction tuning (Zhang et al., 2023; Ouyang et al., 2022; Chung et al., 2024; Zheng et al., 2023) is a strategy commonly adopted in modern LLM training to enhance model generalization by exposing models to various prompts. In the realm of multimodal large language models (MLLMs), visual instruction tuning (Liu et al., 2024c) has significantly improved their instruction-following capabilities when processing multimodal data. This process typically involves two stages: the first stage trains an adapter between the visual encoder and the LLM using image captioning data; in the second stage, the LLM and the adapter are jointly trained with instruction-following data that encompasses multiple tasks in a question-answer format. While previous MLLMs have primarily focused on text generation, recent research is exploring the use of LLMs for representation learning. Specifically, E5-Mistral (Wang et al., 2023) leverages LLMs as embedding models by training them on various retrieval tasks specified by instructions. E5-V (Jiang et al., 2024) extends this approach to multimodal domains; however, its training remains based on pure text pairs, and the full potential of MLLMs for multimodal embeddings is not fully realized. In this paper, we propose a novel approach to train an instruction-aware model that generates multimodal embeddings through two stages: embedding alignment and instruction contrastive learning.

### A.2    COMPOSED IMAGE RETRIEVAL

Composed Image Retrieval (CIR) involves finding images related to a source image under a specified condition, typically provided as a modifier text. This task has practical applications in e-commerce, recommendation systems, and more. Due to the difficulty of acquiring specific datasets for various CIR tasks, recent research has focused on Zero-Shot CIR (ZS-CIR). Previous methods primarily represent the reference image as specific tokens and concatenate them with text tokens for retrieval (Saito et al., 2023; Karthik et al., 2023; Tang et al., 2024; Suo et al., 2024; Agnolucci et al., 2024; Gu et al.). With the advent of Multimodal Large Language Models (MLLMs), researchers have begun incorporating LLMs into this domain. For instance, CIReVL (Karthik et al., 2023) leverages two MLLMs: one for generating image captions and another for combining captions with modifier texts for retrieval. FROMAGe (Koh et al., 2023) and MCL (Li et al.) explore using LLMs for embeddings, but the LLMs are mainly used as text encoders. Despite the rapid development of MLLMs exhibiting strong generalization, instruction-following, and zero-shot capabilities in multimodal data, their applications to CIR tasks are rarely explored. In this paper, we leverage MLLMs as embedding models for CIR tasks, enabling direct encoding of images and modifier texts within a single model.

## B    TRIPLET DATA GENERATION

### B.1    DATA PROCESSING

We utilize GPT-4o (Achiam et al., 2023) to process and generate triplet data. Given an image and its caption, we use the caption as a prompt to GPT, which then derives the modifier text and the modified caption. The detailed prompt structure is shown in Figure 6. Specifically, the prompt is divided into three parts: task definition, requirements, and few-shot examples.

Our data generation process differs from MCL (Li et al.) in several aspects. First, we leverage GPT-4o (Achiam et al., 2023) instead of LLAMA2 (Touvron et al., 2023), allowing for more generalizable and creative content generation. Second, GPT-4o has a larger context window, enabling us to incorporate more complex techniques within the prompt. Unlike MCL, which directly presents the output modifier text and corresponding caption in few-shot examples, we divide the generation process into several steps using the Chain of Thought method (Wei et al., 2022). We instruct GPT to first identify key points in the example caption, then selectively alter some of them as modifications, and finally derive the modified caption. This step-by-step generation ensures that the generated modifier text and corresponding caption are reasonable and closely related to the original caption. *At the time the major work of this paper is finished, the MCL dataset has not been released. We will defer the comparison between two datasets in the future work.*

Our pipeline differs from the training set derivation in (Vaze et al., 2023). While they use text scene graphs to identify subjects, predicates, and objects, their modifier instruction is generated by simply replacing one element with another concept from the dataset, leading to limited creativity and diversity.

I am creating a multi-modal dataset for Composed Image Retrieval (CIR). The goal is to generate pairs of source and target images, along with a modification instruction that describes how to transform the source image into the target image.

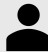

Your Task:
1. Input: I will provide you with a source image caption.
2. Instruction Generation: Brainstorm a modification instruction based on the source caption. This instruction should be a clear, concise description of a plausible change that can be applied to the source image.
3. Modified Caption Generation: Apply the modification instruction to the source caption to create a modified caption that describes the target image after the change.
4. You should output the modification instruction and modified caption only.

Requirements:
1. The modification instruction should focus on a single, significant change (e.g., changing an object's color, altering the setting, modifying an action).
2. The modified caption should reflect only the changes specified in the instruction while keeping the rest of the description consistent with the source caption.
3. Ensure that the instruction and modified caption are coherent and plausible.

Example #1:
Input:
Source Caption: A Husky is lying on the grass.
**Brainstorming**:
The caption contains an object husky, an action lying, and a background grass. One plausible change is altering the action of the dog from lying to running. The modified caption then becomes: a husky is running on the grass.
Output:
Modification Instruction: The dog is running.
Modified Caption: A husky is running on the grass.

Example #2: ……

Input:
Source Caption: a very typical bus station

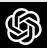

Brainstorming:
The caption describes a location, "a very typical bus station". One significant change could be altering the time of day, which affects the lighting and activity at the location. Transitioning from day to night can introduce new elements like artificial lighting and perhaps a quieter atmosphere.

Output:
Modification Instruction: Change the time of day to night.
Modified Caption: A very typical bus station at night.

Figure 6: We prompt GPT-4o to generate triplet data from CC3M. Our prompt consists of three parts: the first part (orange) defines the task we aim to complete; the second part (blue and purple) specifies the details and requirements of the task; and the third part (black) provides examples for triplet generation, where the modifier text is brainstormed step by step. The key concepts in the captioned are identified and subsequently selected concepts are altered. The modified caption is derived accordingly. Finally, we provide the input (red). GPT then outputs the modifier text and the corresponding caption based on the query caption (green).

## B.2 Data Details

After filtering invalid images and failed prompts, we acquire the CC3M-Instruct dataset with 2M triplets. Triplet examples are shown in Figure 7.

## C Prompt Templates

Templates for training are shown in Table 7.

### C.1 Templates for Training

| Task | Instruction Template |
|---|---|
| Image Modification | <Image> The image is conditioned on the following prompt: {modifier text}, summarize the image and the prompt to retrieve a description of the image changed by the condition: 
 <Image> Given the image conditioned by the prompt: {modifier text}, condense the essence of the image and the prompt into a single word to fetch a description of the altered image: 
 <Image> Using the prompt to condition the image: {modifier text}, provide one word that encapsulates the overall concept of the conditioned image to retrieve its description: 
 <Image> Based on the image influenced by this prompt: {modifier text}, distill the description of the conditioned image and the prompt into one word to access the altered description: 
 <Image> With the image modified according to the prompt: {modifier text}, summarize both the image and the prompt to obtain a description of the conditioned image: 
 <Image> Condition the image with this condition: {modifier text}. Summarize the result: 
 <Image> Using this prompt: {modifier text}, describe the conditioned image: 
 <Image> Apply the prompt: {modifier text} to the image. Provide one word for the conditioned image: 
 <Image> Given this prompt: {modifier text}, condense the conditioned image into one word: 
 <Image> {modifier text}: |
| Image Summary | <Image> Summary: 
 <Image> Caption: 
 <Image> Summarize the image for retrieval: 
 <Image> A short image caption: 
 <Image> A short image description: 
 <Image> Provide a description of what is presented in the photo: 
 <Image> Please provide a short depiction of the picture: 
 <Image> Using language, provide a short account of the image: 
 <Image> Use a word to illustrate what is happening in the picture: |
| Caption Summary | <Caption> Summary: 
 <Caption> Summarize the caption for retrieval: 
 <Caption> A shorter description is: 
 <Caption> Shorter caption: 
 <Caption> "" |

Table 7: Instruction templates for different tasks. In **Image Modification**, the modifier text combined with the selected template serves as the formatted prompt. **Image** and **Caption Summary** instruct the model to generate a global representation for images or captions.

### C.2 Templates for Zero-shot Inference

*CIRR & CIRCO*

**Image Captioning**

<Image> Describe this image in one word:

**Image Modification**

<Image> Modify this image with {modifier text}, describe the modified image in one word:

*FashionIQ*

**Image Captioning**

<Image> Describe this {data type in fashioniq} in one word based on its style:

**Image Modification**

<Image> Modify the style of this {data type in fashioniq} based on {modifier text}. describe this modified {data type in fashioniq} in one word based on its style:

*GeneCIS*

**Image Captioning**

<Image> Summarize the image for retrieval:

**Image Modification**

<Image> Describe the image in one word with a specific focus on the attribute {specific attribute}:

<Image> Describe the image in one word with a specific change of the attribute {specific attribute}:

<Image> Describe the image in one word with a specific focus on the object {specific object}:

<Image> Describe the image in one word with a specific change of on the object {specific object}:

# D TRAINING DETAILS

## D.1 MLLM TRAINING

We use the code and data from xtuner/llava-phi-3-mini-hf (Contributors, 2023) to train a variant of LLaVA-Phi. *Note that the goal of this step is solely to make our experiments consistent with the baselines.* Section 3.4.3 has demonstrated that our training strategy can be directly applied to existing MLLMs. The checkpoint of the variant LLaVA-Phi will also be released for reproducibility. MLLM training and model details are provided as follows.

| Config | Value |
|---|---|
| Visual Encoder | openai/clip-vit-large-patch14 |
| Image Resolution | 224x224 |
| Language Model | microsoft/Phi-3.5-mini-instruct |
| Adapter | MLP |
| Pretraining Strategy | Frozen LLM, Frozen ViT |
| Fine-tuning Strategy | Full LLM, Full ViT |
| Pretrain Dataset | ShareGPT4V-PT (1246K) (Chen et al., 2023) |
| Fine-tune Dataset | InternVL-SFT (1268K) (Chen et al., 2024) |
| Pretrain Epoch | 1 |
| Fine-tune Epoch | 2 |

Table 8: Configurations of Training LLaVA-Phi

| | xtuner/llava-phi-3-mini-hf | microsoft/Phi-3.5-vision-instruct | E5-V |
|---|---|---|---|
| Size | 4.14B | 4.15B | 8.35B |

Table 9: Number of parameters of different models

## D.2 INSTRUCTCIR TRAINING

Detailed training configs are shown in Table 10.

| Training Config | Value |
|---|---|
| DeepSpeed | ZeRO-2 |
| LoRA R | 64 |
| LoRA Alpha | 16 |
| Model Max Length | 512 |
| Precision | FP16 |
| Epochs for both stages | 1 |
| Batch Size Per GPU in Stage 1 | 48 |
| Batch Size Per GPU in Stage 2 | 64 |
| Gradient Accumulation Steps | 1 |
| Learning Rate | 2E-05 |
| Weight Decay | 0 |
| Warm Up Ratio | 0.03 |
| LR Scheduler Type | Cosine |

Table 10: Configurations of Training InstructCIR.

## E MORE EXPERIMENT RESULTS

Table 11, 12, 13 demonstrate the complete results of InstructCIR that is trained with LLaVA-Pretrain (Liu et al., 2024c) only in the first training stage.

| Method | CIRR | | | | | | CIRCO | | | |
|---|---|---|---|---|---|---|---|---|---|---|
| | R@1 | R@5 | R@10 | $R_s$@1 | $R_s$@2 | $R_s$@3 | mAP@5 | mAP@10 | mAP@25 | mAP@50 |
| InstructCIR$_{lp}$ | 35.08 | **65.25** | 76.53 | 67.52 | 84.13 | 92.08 | 22.19 | 23.62 | 26.01 | 27.20 |
| InstructCIR$_{full}$ | **35.18** | 65.12 | **77.61** | **67.54** | **84.77** | **93.61** | **22.32** | **23.80** | **26.25** | **27.32** |

Table 11: **Comparison of Zero-Shot CIR Models on CIRCO and CIRR Test Sets.** InstructCIR$_{lp}$ refers to InstructCIR that is trained with LLaVA-Pretrain only in the first training stage. InstructCIR$_{full}$ is trained with both LLaVA-Pretrain and FOIL in the first training stage.

| Method | Shirt | | Dress | | Toptee | | Average | |
|---|---|---|---|---|---|---|---|---|
| | R@10 | R@50 | R@10 | R@50 | R@10 | R@50 | R@10 | R@50 |
| InstructCIR$_{lp}$ | 29.85 | 49.98 | 25.04 | 45.60 | 31.74 | 53.26 | 28.90 | 49.61 |
| InstructCIR$_{full}$ | **30.96** | **50.10** | **25.11** | **46.18** | **32.32** | **54.22** | **29.46** | **50.16** |

Table 12: **Comparison of Zero-Shot CIR Models on FashionIQ**. InstructCIR$_{lp}$ refers to Instruct-CIR that is trained with LLaVA-Pretrain only in the first training stage. InstructCIR$_{full}$ is trained with both LLaVA-Pretrain and FOIL in the first training stage.

| Method | Focus Attribute | | | Change Attribute | | | Focus Object | | | Change Object | | | Average |
|---|---|---|---|---|---|---|---|---|---|---|---|---|---|
| | R@1 | R@2 | R@3 | R@1 | R@2 | R@3 | R@1 | R@2 | R@3 | R@1 | R@2 | R@3 | R@1 |
| InstructCIR$_{lp}$ | 20.35 | 33.35 | 45.05 | 15.39 | 28.39 | 37.69 | 16.58 | 26.69 | **37.19** | **17.14** | 27.86 | **38.62** | 17.37 |
| InstructCIR$_{full}$ | **21.25** | **34.55** | **46.85** | **16.15** | **28.74** | **39.73** | **17.55** | **28.01** | 36.94 | 17.04 | **28.98** | 37.70 | **18.00** |

Table 13: **Comparison of Zero-Shot CIR Models on GeneCIS**. InstructCIR$_{lp}$ refers to InstructCIR that is trained with LLaVA-Pretrain only in the first training stage. InstructCIR$_{full}$ is trained with both LLaVA-Pretrain and FOIL in the first training stage.

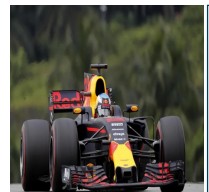

**Modification Instruction:**
The racecar is now a futuristic hovercraft.

**Modified Caption:**
Racecar driver steers his futuristic hovercraft during video game subject.

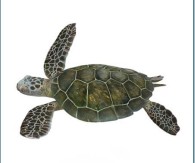

**Modification Instruction:**
The turtle is swimming in a coral reef.

**Modified Caption:**
Green sea turtle swimming in a vibrant coral reef.

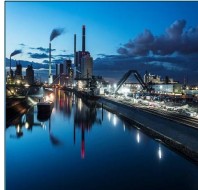

**Modification Instruction:**
Include a full moon in the sky.

**Modified Caption:**
Industrial plants in the distance at night under a full moon in the sky.

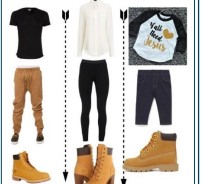

**Modification Instruction:**
Change the boots to sneakers.

**Modified Caption:**
A fashion look featuring blouses, a pair of leggings, and sneakers.

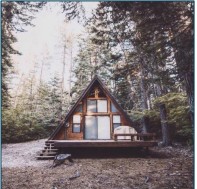

**Modification Instruction:**
Describe the cottage during winter.

**Modified Caption:**
A cottage in the picturesque village covered in snow during winter.

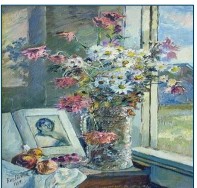

**Modification Instruction:**
The flowers are replaced with a small potted cactus.

**Modified Caption:**
Vase with a small potted cactus and book by the window.

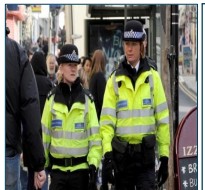

**Modification Instruction:**
During a rainy night.

**Modified Caption:**
Police officers were highly visible on the streets during a rainy night at the weekend.

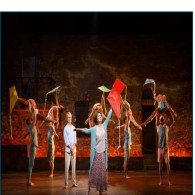

**Modification Instruction:**
Focus on the dancer performing a solo act on stage.

**Modified Caption:**
The dancer performing a solo act on stage, separate from the cast in the vignette.

Figure 7: Triplet Examples from CC3M-Instruct

