# OpenReview forum: "Instruction Contrastive Tuning for Zero-shot Composed Image Retrieval"
_ICLR.cc/2025/Conference — ICLR 2025 Conference Withdrawn Submission_

### Official Review · Reviewer_uwfn · 2024-10-22

**Soundness:** 3
**Presentation:** 3
**Contribution:** 2
**Rating:** 5
**Confidence:** 3

**Summary:**

This paper addresses the challenges of Composed Image Retrieval (CIR), where the goal is to retrieve a target image based on a reference image and a textual modification. The approach involves a two-stage training strategy includes Embedding Alignment: The MLLM is trained on image-caption pairs to transform its text generation capabilities into effective embedding extraction; and the Instruction-Aware Tuning: Using a triplet dataset derived from image-caption data, the model is fine-tuned to generate embeddings that align with the combined image and modification text. This method allows InstructCIR to excel in Zero-Shot CIR (ZS-CIR), showing significant improvements over state-of-the-art methods on benchmarks like FashionIQ, CIRR, CIRCO, and GeneCIS. The paper demonstrates the potential of MLLMs in CIR tasks, especially in understanding and following modification instructions.

**Strengths:**

A new embedding strategy based on instruction-tuned MLLMs.

A two-stage training strategy that can transforms an MLLM’s strong text generation capabilities to effective embedding extraction.

Strong results on four datasets:  FashionIQ, CIRR, GeneCIS, and CIRCO compared with many other baselines.

**Weaknesses:**

The novelty of this paper is limited. There are already many existing works on image/text retrieval with MLLMs and finetuning MLLMs with instruction tuning data. [1][2][3] This paper is more like a combing of these two.

[1] LightningDOT: Pre-training Visual-Semantic Embeddings for Real-Time Image-Text Retrieval
[2] Position-Enhanced Visual Instruction Tuning for Multimodal Large Language Models
[3] ComCLIP: Training-Free Compositional Image and Text Matching
[4] Generative Cross-Modal Retrieval: Memorizing Images in Multimodal Language Models for Retrieval and Beyond

Using GPT4o to process and generate triplet data may bring additional biases. Additional evaluation such as human evaluation is needed to validated the approach. For example, asking humans to rate the quality or relevance of a sample of the generated triplets, or compare them to manually created triplets or trained a discriminator to filter biased data.

The comparison between different methods may be not fair as they also needed to finetuned on the datasets, e.g. CIReVL, MCL.

**Questions:**

Can the proposed two-strage training pipeline applied to other baseline models?

The potential scaling law for the propsed approach. For example, when the llm backbone is replaced with larger phi models.

---

> ### Author Response · Authors · 2024-11-15
>
> Thank you for your valuable review.
>
> For the novelty concern, please refer to the General Response.
>
> **Data Generation Quality**
>
> In our data generation process, we employ an automatic filtering strategy: outputs that fail to conform to specific templates for modification instructions and captions are discarded. We also leverage the Chain of Thoughts approach to enhance the quality of triplet generation. This strategy is labor-free and easy to implement, making the auto-generated datasets valuable resources for training, as outlined in multiple existing works.
>
> **Other Baselines**
>
> CIReVL is a training-free approach and inherently non-trainable. MCL, on the other hand, leverages similar triplet datasets based on CC3M. However, at the time of our experiments, MCL’s training codes were not yet available.

---

### Official Review · Reviewer_jUcP · 2024-11-01

**Soundness:** 2
**Presentation:** 2
**Contribution:** 2
**Rating:** 3
**Confidence:** 4

**Summary:**

This paper introduces a two-stage training strategy for instruction-tuning a Multimodal Large Language Model (MLLM) for Zero-Shot Composed Image Retrieval (ZS-CIR). Training by this strategy, this paper proposed InstructCIR, an embedding method leveraging an instruction-tuned MLLM to generate composed embedding for composed image retrieval. InstructCIR demonstrates outperforming state-of-the-art baselines across four public datasets.

**Strengths:**

1.	Instruction-tuning an MLLM for CIR is interesting.
2.	The various ablation studies show the effectiveness of InstructCIR.
3.    The proposed method demonstrates outperforming state-of-the-art baselines across four public datasets.

**Weaknesses:**

1. The setting of this paper is inconsistent with standard ZS-CIR tasks [1,2,3,4,5,6,7], which aim to leverage the frozen CLIP model with alignment knowledge to achieve zero-shot compose image retrieval. However, this method of pretraining the entire MLLM diverges from the established settings of published ZS-CIR models, which may not be a "Zero-Shot" setting in the ZS-CIR task. Therefore, it is unfair to compare. It seems more aligned with what "the Semi-Supervision CIR" [8] aims to address. This inconsistent setting may cause potential data leakage in the instruction tuning process, which is a fitting bias of the CIR data, leading to potential data leakage. Moreover, this method required training of the entire MLLM, introducing significant training parameter size, computational resources cost, and time increase, which is unfair compared to existing ZS-CIR methods.

2. The novelty is confused. The primary motivation :``these base models are not trained on instruction-aware data, limiting their ability to effectively combine visual and textual cues.’’ is unreasonable to me. The authors should carefully explain the relation between “instruction-aware data” and “ZS-CIR task”; for example, why does ZS-CIR require instruction-aware data?

3. The technology contribution is limited. The two-stage training method is really similar to LLaVA [9], which only add a special token [EOS] as the global feature for retrieval. The instruction data generation process, which leverages GPT-4o, is not analyzed. The qualitative results may show that GPT-4o could achieve the goal of this method, which aims to generate the modified caption. Therefore, I would like to question the necessity of this method.

4. The inference stage for retrieval is NOT included in this paper, and the code are not given. This paper only shows the two-stage training process. However, the retrieval process, which is one of the most important stages, may be overlooked. The retrieval seems to leverage the MLLM encoder for all candidate images first and then uses the compose embedding as a query for retrieval. In this way, the computational resources cost and time required by this method are challenging for CIR, which is used in online e-commerce and recommendation systems. This raises me concerns about the practical significance of this method.

5. The comparison of the main results might not be complete. For example, the SoTA of the CIRCO dataset is CIReVL(ViT-G-14) [6], and CIRR is LinCIR (ViT-G-14) [1], which this paper does not compare. This method includes a significant increment of modal size and pre-trained knowledge, which needs to be compared with ZS-CIR methods with larger CLIP backbones.

6. Need more ablation studies. For example, what is the influence of other MLLM? What is the influence of GPT-4 or GPT-4-mini in data generation process?

Reference

[1] Geonmo Gu, Sanghyuk Chun, Wonjae Kim, Yoohoon Kang, and Sangdoo Yun. Language-only efficient training of zero-shot composed image retrieval. In CVPR, 2024.

[2] Kuniaki Saito, Kihyuk Sohn, Xiang Zhang, Chun-Liang Li, Chen-Yu Lee, Kate Saenko, and Tomas Pfister. Pic2word: Mapping pictures to words for zero-shot composed image retrieval. In CVPR, 2023.

[3] Suo Y, Ma F, Zhu L, et al. Knowledge-enhanced dual-stream zero-shot composed image retrieval[C]//Proceedings of the IEEE/CVF Conference on Computer Vision and Pattern Recognition. 2024: 26951-26962.

[4] Tang Y, Yu J, Gai K, et al. Context-I2W: Mapping Images to Context-dependent Words for Accurate Zero-Shot Composed Image Retrieval[C]//Proceedings of the AAAI Conference on Artificial Intelligence. 2024, 38(6): 5180-5188.

[5] Du Y, Wang M, Zhou W, et al. Image2Sentence based Asymmetrical Zero-shot Composed Image Retrieval[J]. ICLR 2024.

[6] Karthik S, Roth K, Mancini M, et al. Vision-by-language for training-free compositional image retrieval[J]. ICLR 2024.

[7] Alberto Baldrati, Lorenzo Agnolucci, Marco Bertini, and Alberto Del Bimbo. Zero-shot composed image retrieval with textual inversion. In ICCV, 2023.

[8] Jang Y K, Kim D, Meng Z, et al. Visual Delta Generator with Large Multi-modal Models for Semi-supervised Composed Image Retrieval[C]//Proceedings of the IEEE/CVF Conference on Computer Vision and Pattern Recognition. 2024: 16805-16814.

[9] Haotian Liu, Chunyuan Li, Qingyang Wu, and Yong Jae Lee. Visual instruction tuning. Advances in neural information processing systems, 36, 2024c.

**Questions:**

1.	Why ZS-CIR requires instruction-aware data?
2.	Why just leverage GPT-4o to achieve your goal?
3.	What is necessary for this method?
4.	How did you conduct your retrieval process?
5.	What do training parameter sizes compare to the existing ZS-CIR (e.g., Pic2Word)?
6.	What is the influence of different MLLMs?
7.	What is the influence of GPT-4 or GPT-4-mini in the data generation process?

---

> ### Author Response · Authors · 2024-11-15
>
> Thank you for your valuable review.
>
> **Inconsistent Baselines**
>
> ZS-CIR is a type of problem and is model-agnostic. Therefore, frozen CLIP models are not the only choice for tackling these problems. Several approaches compared in our experiments, including E5-V, CIReVL, FROMAGe, and MCL, also involve LLMs, and all are ZS-CIR models. In contrast, approaches like VDG are semi-supervised methods, as their primary approach involves annotating data with MLLMs rather than directly conducting ZS-CIR. Additionally, VDG requires different data annotations for specific downstream tasks. In practice, specific data annotations may not always be available. InstructCIR, however, is trained once and directly applied to various scenarios, maintaining the zero-shot setting.
>
> **Confusing Novelty**
>
> We clarify that CIR is inherently an instruction-following task, differing from traditional image-text retrieval based on semantic similarity. CIR requires the model to understand what changes are necessary according to the instructions and apply them to the image. For example, if the instruction is “change the dog to a cat in the photo,” the model must comprehend the semantics to replace the dog object with a cat. Therefore, instruction-following capability is essential, making an instruction-tuned MLLM an ideal embedding model. However, directly using MLLMs presents a challenge, as they are primarily designed for text generation rather than embedding extraction. Our two-stage training strategy effectively addresses this by enabling the MLLM to learn a joint multimodal embedding space and fine-tune it with triplet data similar to the CIR format. These strategies significantly improve performance compared to baselines in experiments.
>
> **Limited Technical Contribution**
>
> While our approach draws inspiration from LLaVA, InstructCIR is fundamentally different. LLaVA is designed for text generation, whereas our approach focuses on composed image retrieval. InstructCIR is an embedding model rather than a generation model like LLaVA. Furthermore, our training strategy is specifically curated for composed image retrieval, aiming to learn a joint embedding space and adapt the model for instruction-aware composed embeddings. In contrast, LLaVA’s training strategy primarily adapts an LLM for multimodality, enabling it to become an MLLM.
>
> Regarding directly using ChatGPT for CIR, one of our baselines, CIReVL, leverages this approach but requires BLIP2 for captioning during inference. Our model is significantly smaller and faster than CIReVL while achieving better performance. Additionally, ChatGPT may not be feasible in certain commercial scenarios due to privacy concerns.
>
> **Resource and Time Consumption in Practical Applications**
>
> Our approach leverages Phi-3.5-mini as the LLM backbone, a much smaller model than general LLMs, enabling deployment even on mobile devices. Although larger than previous methods like Pic2Word, InstructCIR is the smallest among LLM-based approaches (FROMAGe, CIReVL, MCL). In practical applications, target image embeddings can be pre-computed and cached, further reducing retrieval time.
>
> **Comparison Not Fair?**
>
> We ensured a fair comparison by retraining a variant of ViT-L with an image size of 224x224, rather than using the commonly employed size of 336x336 in MLLMs, to align with baseline models. CIR is an image retrieval task, making the image component critical for consistency. We also note that baselines like MCL use LLMs and ViT-L while maintaining comparisons with ViT-L in their original papers.
>
> **Data Processing with GPT-4 and GPT-4-mini**
>
> The primary goal of generating the triplet dataset is to support the second training stage, and it is not listed as a contribution. Analyzing which LLM generates the best triplet data is outside the scope of this paper. Given time and budget constraints, we selected GPT-4o for its superior overall performance and believe the dataset generated is of high quality.

---

### Official Review · Reviewer_Mj5A · 2024-11-03

**Soundness:** 2
**Presentation:** 2
**Contribution:** 2
**Rating:** 3
**Confidence:** 4

**Summary:**

This paper focuses on Zero-Shot Composed Image Retrieval (ZS-CIR). To effectively combine visual and textual cues, the authors propose an embedding method utilizing an instruction-tuned Multimodal Large Language Model (MLLM) to generate unified embeddings that integrate images and modification instructions. They introduce a two-stage training strategy to efficiently transform the MLLM’s text generation capabilities into embedding extraction, and further refining its ability to follow modification instructions in CIR. Experiments on four datasets shows the effectiveness of the proposed method, compared to the selected baselines.

**Strengths:**

1. The authors propose a new method based on MLLM.
2. The fine-tuning procedure of the proposed method is rational.
3. The writing is easy to follow.

**Weaknesses:**

1. Data leakage: the data in the first training stage of the proposed method utilize the data source of the adopted CIR datasets, such as MSCOCO. In zero-shot settings, no data (no matter training data or test data) in the target dataset can be utilized in model construction or optimization. Section 3.2 has not addressed this problem.
2. Important baselines are missing: As the authors utilize GPT-4o to combine the image and the modification text into a combined text, an important baseline is directly utilizing the combined text to retrieve the composed image using image-text retrieval methods, such as CLIP. This is because this baseline is a simple and direct method, which is very easy to implement in applications.
3. The proposed model is much larger (in terms of memory) and slower (in terms of speed) than the baseline models in the inference. The efficiency is concerned.

**Questions:**

Please respond to Weaknesses.

---

> ### Author Response · Authors · 2024-11-15
>
> Thank you for your valuable review.
>
> **Data Leakage**
>
> We pay careful attention to ensuring there is no data leakage. Although FOIL uses some images from MSCOCO, it does not overlap with benchmark datasets as they come from different versions and splits. Additionally, FOIL does not include modification instructions used in CIR benchmarks. Therefore, our model does not encounter triplet data used in CIR benchmarks during training.
>
> To further eliminate concerns, we have included results of InstructCIR trained only with LLaVA-Pretrain in Stage 1 in Appendix E. Even without FOIL data, our model significantly outperforms baselines. The purpose of using multiple datasets in Stage 1 is to show the benefits of training with diverse contexts.
>
> **Baselines Missing**
>
> We would like to point out that methods directly using modified captions with retrieval models like CLIP have been verified by CIReVL, which is included in our comparisons. However, as captions are unavailable during inference, CIReVL employs BLIP2 (a much larger MLLM) to generate captions for testing images and then uses ChatGPT to derive modified captions. In experiments, InstructCIR significantly outperforms CIReVL.
>
> **Training Speed**
>
> Please refer to the General Response.

---

### Official Review · Reviewer_71WG · 2024-11-03

**Soundness:** 3
**Presentation:** 2
**Contribution:** 3
**Rating:** 6
**Confidence:** 3

**Summary:**

This paper proposes a two-stage training method that instruction-tunes MLLMs for zero-shot CIR tasks. In the first stage, the authors train the MLLMs to encode images and text into a joint embedding space. In the second stage, the model is trained to generalize to different instructions, enabling better zero-shot CIR performance. Experiments demonstrate results across various benchmarks, along with  ablation studies on different modules of the model.

**Strengths:**

- The model leverages LLMs to generate modified text and target captions, presenting a novel and efficient strategy that avoids additional annotation costs.

-  This method achieves significant improvements across various benchmarks. Comprehensive ablation studies illustrate the contribution of each module of the model.

**Weaknesses:**

- My concern is about the efficiency of using MLLMs for zero-shot CIR tasks. Traditional zero-shot CIR methods are typically very efficient, for example Pic2word, SEARLE, LinCIR, Context-I2W. Could the authors provide a comparison of inference times, like average inference time per image when testing on the same device?

- The authors did not specify inference details. Could the authors provide a detailed description of the inference process after training.

**Questions:**

Please refer to the weakness above. I will carefully review the rebuttal and consider the opinions of the other reviewers to adjust my rating.

---

> ### Author Response · Authors · 2024-11-15
>
> Thank you for your valuable review.
>
> For the efficiency concern, please refer to the General Responsen.

---

### Author Response · Authors · 2024-11-15
**General Response**

We thank all reviewers for their valuable comments.

Below we address two common concerns:

**Efficiency**

We pay specific attention to balancing model efficiency and performance. Specifically, we do not leverage general LLMs as our backbone model. Instead, we use an on-device LLM, Phi-3.5-mini, which is lightweight and deployable even on mobile devices. While our approach is larger than previous methods like Pic2Word, it is the smallest among LLM-based models (CIReVL, MCL, FROMAGe, E5-V). Our model is half the size of these baselines while demonstrating significant advancements. Moreover, with the rapid development of on-device LLMs, the backbone LLM in InstructCIR can benefit from lighter and better LLMs in the future.

**Novelty of the Method**

We would like to highlight the novelty of our approach. While existing works address MLLMs and LLMs in CIR, we are the first to use an instruction-aware MLLM as an embedding model for this task. We tackle the challenge of using an MLLM (originally used for text generation) in CIR with our novel training strategy. Specifically, the first training stage builds a multimodal embedding space for retrieval, while the second stage tunes the model to produce instruction-aware composed embeddings.

---

### Note · Authors · 2024-11-15

I have read and agree with the venue's withdrawal policy on behalf of myself and my co-authors.